# Carbon-Neutral ESG Method Based on PV Energy Generation Prediction Model in Buildings for EV Charging Platform

Guwon Yoon [1], Seunghwan Kim [2], Haneul Shin [2], Keonhee Cho [2], Hyeonwoo Jang [1], Tacklim Lee [2], Myeong-in Choi [2], Byeongkwan Kang [2], Sangmin Park [2], Sanghoon Lee [2], Junhyun Park [2], Hyeyoon Jung [2], Doron Shmilovitz [3] and Sehyun Park [1,2,*]

[1] School of Electrical and Electronics Engineering, Chung-Ang University, Seoul 06974, Republic of Korea; gw1206@cau.ac.kr (G.Y.); gostub123@cau.ac.kr (H.J.)
[2] Department of Intelligent Energy and Industry, Chung-Ang University, Seoul 06974, Republic of Korea; tkftn456@cau.ac.kr (S.K.); shaneul7@cau.ac.kr (H.S.); thckwall@cau.ac.kr (K.C.); tacklim34@cau.ac.kr (T.L.); auddlscjswo@cau.ac.kr (M.-i.C.); byeongkwan@cau.ac.kr (B.K.); motlover@cau.ac.kr (S.P.); leessan0@cau.ac.kr (S.L.); wnsgus7522@cau.ac.kr (J.P.); y1204n@cau.ac.kr (H.J.)
[3] School of Electrical Engineering, Tel Aviv University, Tel Aviv 6997801, Israel; shmilo@tauex.tau.ac.il
* Correspondence: shpark@cau.ac.kr; Tel.: +82-2-822-5338

**Abstract:** Energy prediction models and platforms are being developed to achieve carbon-neutral ESG, transition buildings to renewable energy, and supply sustainable energy to EV charging infrastructure. Despite numerous studies on machine learning (ML)-based prediction models for photovoltaic (PV) energy, integrating models with carbon emission analysis and an electric vehicle (EV) charging platform remains challenging. To overcome this, we propose a building-specific long short-term memory (LSTM) prediction model for PV energy supply. This model simulates the integration of EV charging platforms and offer solutions for carbon reduction. Integrating a PV energy prediction model within buildings and EV charging platforms using ICT is crucial to achieve renewable energy transition and carbon neutrality. The ML model uses data from various perspectives to derive operational strategies for energy supply to the grid. Additionally, simulations explore the integration of PV-EV charging infrastructure, EV charging control based on energy, and mechanisms for sharing energy, promoting eco-friendly charging. By comparing carbon emissions from fossil-fuel-based sources with PV energy sources, we analyze the reduction in carbon emission effects, providing a comprehensive understanding of carbon reduction and energy transition through energy prediction. In the future, we aim to secure economic viability in the building energy infrastructure market and establish a carbon-neutral city by providing a stable energy supply to buildings and EV charging infrastructure. Through ongoing research on specialized models tailored to the unique characteristics of energy domains within buildings, we aim to contribute to the resolution of inter-regional energy supply challenges and the achievement of carbon reduction.

**Keywords:** ESG; carbon neutral; renewable energy transition; AI prediction model; data analysis; sustainable building; electric vehicle charging platform

## 1. Introduction

In a special report by the Intergovernmental Panel on Climate Change (IPCC), a plan was presented to achieve carbon neutrality by 2030 based on the carbon emission pathway of global warming, owing to which the planet's temperature is expected to be 1.5 °C higher than that before industrialization, and a decarbonization scenario based on the use of renewable energy. To realize carbon neutrality, understanding the energy consumed in the building sector, which is more than that in other sectors, such as industry, agriculture, and commerce, is pivotal [1]. Owing to rapid population growth worldwide, the demand for residential buildings has increased significantly. Buildings account for 36% of global energy consumption, and energy use in buildings plays an important role in the process

of converting renewable energy [2]. Installation of renewable energy infrastructure has been a major consideration in building design from the perspective of realizing zero-energy buildings and zero-carbon buildings with "zero" carbon emissions [3]. Accordingly, the energy use of buildings is one of the greatest contributors to carbon emissions, and it is urgent to decarbonize buildings in order to design low-carbon cities and operate low-carbon energy infrastructure [4]. To enable a sustainable smart energy city (SSEC) from an energy infrastructure perspective, the integration of city-specific services and businesses is considered to provide benefits to citizens, such as convenience, safety, and cost savings [5]. Therefore, artificial intelligence (AI) is applied to building energy management systems (BEMS) to implement intelligent energy optimization based on the latest information and communications technology (ICT) [6]. Significant efforts have been dedicated to the design of low-carbon buildings, including improving energy efficiency, using renewable energy, and reducing fossil fuel consumption [7]. Energy consumption is increasing by 2% worldwide every year, and total energy production depends heavily on fossil fuels, such as natural gas, coal, and oil, which increases carbon emissions worldwide [8,9]. Energy generation from fossil fuels is considered a major life-threatening factor because of the associated environmental risks and potential for energy crises owing to a decrease in energy resources and an increase in environmental pollution [10–12]. Therefore, by establishing a global strategy for carbon reduction and utilizing renewable energy on a large scale, one can not only solve the carbon emissions problem but also facilitate the switch from fossil fuels to renewable energy to mitigate the impact of climate change [13,14]. In addition, efforts are being made to strengthen carbon neutrality goals and global competitiveness by setting a direction for global strategy and management centered on environment, social, and corporate governance (ESG) for sustainable development. ESG is essential because it leads to sustainability by considering the social and environmental impacts of all categories of industries that are directly or indirectly related to a company.

Photovoltaic (PV) energy generation is growing at a relatively fast rate every year. However, because of the randomness of light and periodicity of day and night, PV energy generation is essentially uncontrollable and highly volatile [15]. Global agreements emphasizing the importance of generating eco-friendly energy focus on PV energy [16]. The usage and conversion methods of sustainable renewable energy sources (RES) are being studied for the same purpose of energy generation that has replaced fossil fuels [17]. In 2019, PV energy supplied 35% of the total energy of renewable energy in the European Union, providing more energy than fossil fuels. According to a forecast by the U.S. Energy Information Administration (EIA), global renewable energy generation is expected to reach 49% in 2050 and increase by 3.6% annually. The SunShot Vision Study conducted in the United States reported that by 2030, 14% of the energy consumed in the United States will be PV energy, with the value expected to double to 27%.

Accurate prediction of energy generation is important to ensure safe operation and economic integration of PV energy [18,19]. Reliable prediction of PV energy generation can reduce the uncertainty surrounding the planning, management, and operation of energy systems [8]. In addition, prediction of PV energy generation can be very helpful to realize balanced operation of renewable energy in the energy market, facilitating the establishment of a stable energy supply system and benefitting users [20]. According to International Energy Agency's Global Electric Vehicle Outlook 2023, new electric vehicle (EV) sales, which stood at about 6.6 million in 2021, will reach 18 million in 2025 and 30 million in 2030, with EVs accounting for 10% of all vehicles in 2030. According to TechSci Research's Global Electric Vehicle Market in 2020, the global EV market is expected to grow at an average annual rate of 21.2%. Therefore, PV energy generation prediction is an important factor in stabilizing the energy supply system and balancing energy usage/supply at the city and building levels [21]. Energy sharing simulation is also important to ensure energy stability, assuming that future EV and charging stations are widely distributed [1].

This study aims to address climate change by pursuing carbon-neutral ESG, transitioning buildings to renewable energy sources, and supplying renewable energy to



PV-energy-based carbon reduction and EV charging infrastructure. As such, a machine learning (ML)-based PV energy generation prediction model is difficult to expand by linking EV charging platforms with a model that analyzes carbon emissions based on PV energy generation prediction models, although many studies have been conducted. Therefore, in terms of renewable energy conversion by applying ML prediction models for each building unit, we write about ways to link EV charging platforms and reduce carbon emissions. PV energy generation prediction using the long short-term memory (LSTM) model plays a very important role in the operation and management of PV plants, and accurate energy generation prediction has a considerable influence on the operation of the energy grid and the establishment of a plant's optimal operation plan. Linking EV charging platforms and PV energy generation prediction analysis plays an important role in pursuing sustainable energy and carbon neutrality; therefore, we analyze the carbon emissions o future renewable energy supplies. Finally, with only data from the area of the building, the location of the sun, and the weather perspective, it is possible to design a carbon reduction model that involves the conversion of renewable energy in the city through PV energy generation simulation from the perspective of urban planning design. In addition, when linking EV charging platforms with the PV energy generation prediction model, EV charging can be adjusted during times when PV energy generation is high in terms of EV charging scheduling to induce direct use of energy produced by PV energy generation. Combining digital innovation technologies to provide data that can be charged with renewable energy through future building infrastructure and EV charging platforms secures economic power in the EV market and eco-friendly charging, supporting the achievement of carbon-neutrality.

The remainder of this paper is organized as follows. A review of the literature is presented in Section 2. Section 3 presents an overall design overview of the proposed simulation model for PV energy prediction. Section 4 describes the implementation and accuracy verification of the proposed PV energy generation prediction model. In Section 5, we explore carbon neutrality based on a prediction model through EV platform linkage. Finally, in Section 6, the expected effects and future prospects of the final proposed model are presented.

## 2. Related Works

In this chapter, we review the relevant literature on the conversion of renewable energy in building units and combination of with EV platform-linked carbon reduction.

### 2.1. PV Energy Generation Prediction Model

To determine the appropriate prediction technique for AI based on the time axis, the collected data are analyzed from the perspective that past patterns in time series data will persist into the future. After analyzing the time series characteristics of the domain, suitable analysis and prediction algorithms are chosen based on the objectives of the prediction and the time axis. The time series data can be characterized by three main patterns. The first pattern is trend variation, which represents a trend tendency and can be depicted as an extension of a straight line or a smooth curve [22]. It either continuously increases or decreases over an extended period or maintains a constant state. The second pattern is irregular variation, which occurs randomly without any predictable regularity [23]. It is not caused by sudden environmental changes or natural disasters but arises unexpectedly. The last pattern is cyclical variation, which shows periodic fluctuations along the trendline over time [24]. It exhibits recurrent changes with a consistent cycle of approximately 2 to 3 years. Based on these cyclic characteristics observed in a significant amount of time series data, an appropriate LSTM prediction model was selected. When there is a scarcity of time series data for the design of a predictive model, it is important to determine the feasibility of implementing a future energy strategy [25]. Owing to their wide prediction range, LSTM models with complex structures are required to predict long-term patterns and dependencies of data. Moreover, large amounts of training data are required, resulting

in increased computational costs and longer model training times [26]. Among AI models, LSTM models are the most accurate for predicting PV energy generation of the standard variety [27,28]. Therefore, the objective of an LSTM-based PV energy generation model is set according to time scale, which is an important factor in the prediction of PV energy generation. Predictive models based on time series data can predict future PV energy generation with superior accuracy [29], and adding large amounts of training data to the input data improves performance [30–33].

As such, other ML algorithms that represent the advantages of LSTM, such as a recurrent neural network (RNN) and a gated recurrent unit (GRU), are compared with LSTM. Compared to LSTM, RNN utilized memory in processing interrelated input data, making it easy to learn data that exhibit specific patterns. However, LSTM models take longer to train and often have lower accuracy compared to traditional RNNs. Despite this, LSTM models excel at modeling time-dependent data, surpassing the capabilities of regular RNNs [34]. Compared to the LSTM algorithm, GRU simplifies the structure, resulting in faster learning but similar performance to that of LSTM. When the amount of data is limited, the parameter-efficient GRU tends to perform better. However, in cases in which there is a large amount of data available, LSTM outperforms GRU in terms of performance [35].

To stabilize energy demand and supply, past weather data, PV energy data, and solar radiation data have been used in predictive models to predict future PV energy generation [19,20]. Weather information related to PV energy generation data are classified into weather grades, and ML is applied to predict the next day's weather [19]. To address the unstable nature of PV energy, researchers have developed a method for predicting PV energy generation using historical data and weather data [36]. In a previous study, we proposed a time series prediction method for each of the ideal and non-ideal weather types [37]. A PV energy generation prediction scheme that can contribute to reducing uncertainty due to variability in weather conditions was proposed in [21]. The efficiency of such predictive models depends on the amount of data, the quality of input data, weather at the data collection site, and the historical solar radiation period being considered [33].

Previous studies have not predicted PV energy based on time series data by linking the location of the sun with the data perspective of weather conditions, only designing a PV energy generation prediction model in single-condition environments. However, the model proposed herein predicts the amount of PV energy generated by linking data from two perspectives.

*2.2. Simulation of PV Energy Generation in Building*

In the context of operating low-carbon buildings, there is an urgent need to introduce effective building energy operation measures and renewable energy conversion technologies [38]. PV energy is a good candidate for use in the architectural sector, which accounts for the largest share of total energy consumption. PV energy utilization in building environments is challenging from several perspectives because of the high density of various rooftop heights and reduced surface area for sunlight exposure on building rooftops [39]. Unexpected losses occur due to shadows of surrounding buildings and clouds [40]. In addition, there are various building facilities on the roof of the building that can cast shadows on solar panels, reducing power production [41]. However, it is convenient to install solar panels on rooftops with limited space because of the uniform and modular characteristics of solar panels [42]. Moreover, the performance characteristics of PV systems installed on building rooftops were evaluated, and a simulation and analysis model for predicting the performance of such systems was developed [43].

Therefore, to increase the amount of PV energy generation in buildings, it is necessary to formulate plans that promote the utilization of renewable energy [44]. The problems of the industry can be addressed by establishing a PV energy demand–supply system based on building rooftop area, building operation and maintenance, energy demand management, and energy purchase [43]. Furthermore, to reduce carbon emissions and

protect the environment, it is crucial to consider stakeholders (homeowners, developers, the private sector, and the government) at all levels when implementing policies and initiatives necessary to build a better future [45].

### 2.3. Renewable Energy Transformation and EV Charging Platform for Carbon Reduction

As urbanization and industrialization accelerate, achieving carbon neutrality and maximum carbon dioxide emissions have become globally shared sustainability goals [46]. Urbanization is a major driver of carbon emissions, making it crucial to analyze the feasibility of applying renewable energy at the building level [17,47]. Renewable energy generation is effective for decarbonizing buildings and quantifying economic impact [4,48]. The transition to renewable energy aims to reduce the greenhouse gas emissions associated with fossil-fuel-based energy generation. Unlike fossil fuels, such as coal, oil, and natural gas, renewable energy sources do not release carbon into the atmosphere, which reduces their carbon footprint substantially. Approaches involving building interventions and government-promoted clean energy initiatives integrate buildings and renewable energy to meet national carbon reduction goals [45]. In energy simulations, optimally configured hybrid systems generate 11.72% of the average calculated carbon emissions of conventional energy systems while providing similar amounts of energy to regular residential buildings and fulfilling 100% of the energy demands of green buildings [49].

The transition to renewable energy and EV platform linkage are crucial for addressing climate change and must be facilitated on an ongoing basis using innovative digital technologies. Many countries worldwide are using PV energy generation systems to reduce carbon emissions [50–52]. Alongside the rapid transition to renewable energy, governments worldwide are striving to move away from fossil fuel systems to forge sustainable development pathways and reduce emissions [39,53,54]. To advance the development of renewable-energy-based EV ecosystems and effectively manage EV charging infrastructure at a high level, flexible charging during daytime is essential. Weather-dependent PV energy generation and energy costs during peak hours are important factors that significantly influence the system design for EV operations [55]. By considering EV charging based on scheduled PV energy generation for each weather scenario, it is possible to reduce energy consumption [56]. Supplying PV energy generated during the daytime to EV charging infrastructure allows for proportional reduction in carbon emissions in line with energy usage [57]. As a result, renewable energy generation methods play a significant role in reducing carbon emissions, and worldwide, concerted efforts are being made to actively expand the use of renewable energy.

## 3. Design of Simulation Model for PV Energy Prediction

In this paper, we propose a simulation model to predict PV energy generation. The proposed model analyzes the correlation between distributed energy management in an energy-centered circular economy and carbon emissions. To this end, we formulate an architecture to guide the design of the simulation model for prediction of PV energy generation, focusing on the direction of the model. In this section, we discuss the design direction of the proposed simulation model for prediction of PV energy generation, data collection and processing methods used herein, the design of the LSTM model, and carbon neutrality based on a predictive model with EV platform linkage.

### 3.1. Architecture of Predictive Simulation Model

Figure 1 presents the architecture of the proposed simulation model for prediction of PV energy generation, which outlines the direction of the simulation approach from the perspective of energy transition to carbon neutrality. To address the limitations of extant research and design an advanced simulation model, a four-layer approach is proposed, as depicted in Figure 1. The direction of the model within Figure 1 is encompassed to derive the final predictions of PV energy generation.

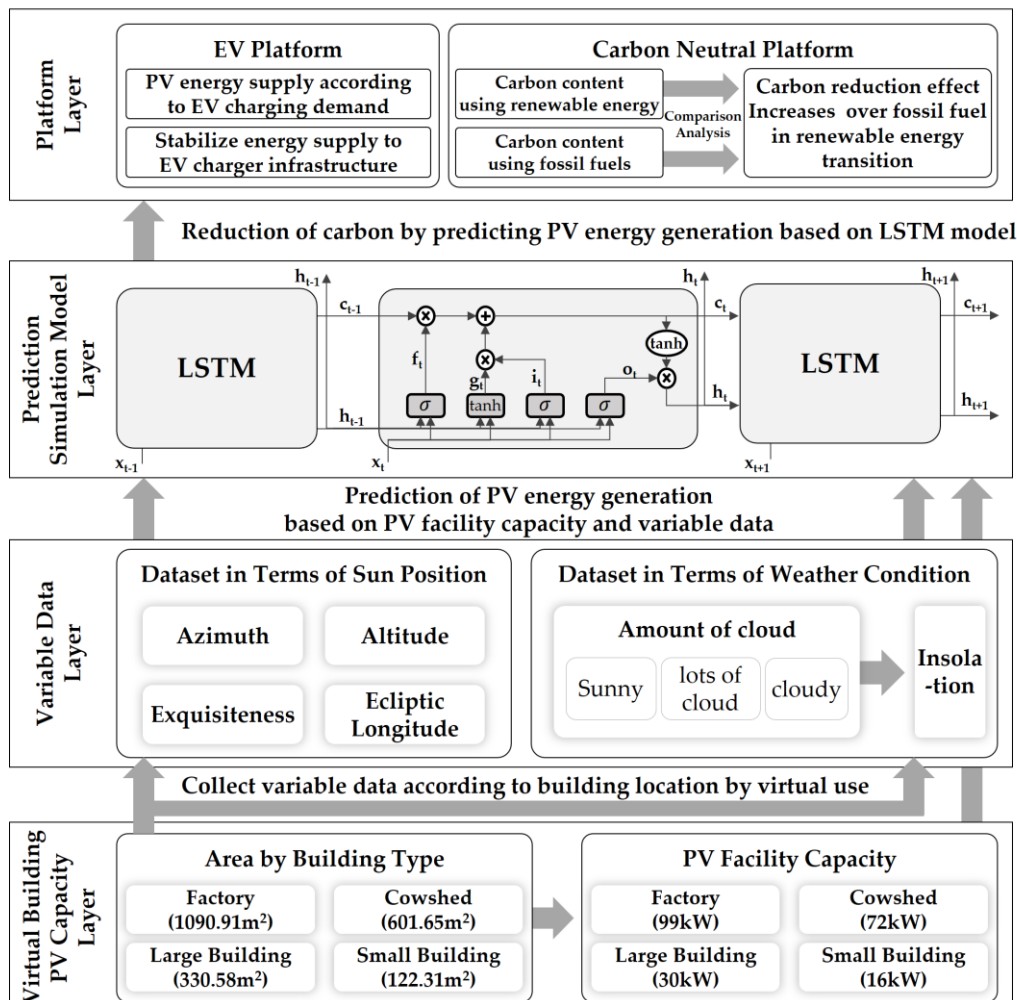

**Figure 1.** Architecture of proposed renewable energy prediction model for carbon neutrality.

The direction can be divided into four areas: data collection, prediction simulation, carbon reduction, and EV platform linkage. In the first area, data collection, building area by usage, equipment capacity, correlated solar position, and weather data are collected by removing missing values. In the second area, the collected correlated data are tracked using ML algorithms to predict PV energy generation. In the third area, the carbon emissions due to conventional energy generation and PV energy generation are calculated to analyze carbon tracking and reduction effects based on the carbon life cycle. Finally, in connection with the EV platform, a plan to supply energy from PV energy generation according to the charging demand in the EV charging infrastructure is presented.

### 3.1.1. Area Selection by Building Type

The selection of building area by building type is determined based on the building's size, usage, characteristics, functions, and number of users. For example, in residential buildings, the appropriate area can be determined by considering factors such as the number of residents, lifestyle and activities, and need for personal and shared spaces. In commercial buildings, the appropriate area can be determined by considering the number of users, types of business and services, and need for shared spaces. In terms of energy efficiency and environmental friendliness, the area of the facilities that can be integrated with PV energy generation is considered when determining the building area. After selecting appropriate areas for each building type using the aforementioned methods, efficient building operation and enhanced carbon reduction effects can be realized. Therefore, herein, a simulation model for the prediction of PV energy generation is proposed

by virtualizing the areas of representative building types with the aim of reducing carbon emissions.

### 3.1.2. Simulation of Energy Transition Perspective for Carbon Reduction

To reduce carbon emissions, the integration of energy generation resources with equipment characteristics and distributed energy systems is analyzed. By using the collected data on energy and management, energy production patterns are analyzed to formulate energy transition simulation approaches. By utilizing the energy data of each building type, renewable energy is defined, and the relationship between energy production and load is predicted using historical data. Considering the importance of the key components of future energy supply networks and future energy solutions, we propose approaches for the development of models to predict renewable energy generation.

The carbon reduction model is applied to PV energy generation equipment and energy production scheduling within buildings to mitigate carbon and particulate matter emissions. Through carbon emission tracking and management, the carbon reduction effects are analyzed, and a low-carbon energy production prediction model is proposed by integrating PV energy generation and data. This model can be used to transition the existing energy supply system and ensure a safe life for citizens in terms of governance and community perspectives.

### 3.2. Construction of Data Linked to PV Energy Generation

PV energy generation refers to the amount of electricity generated by solar panels, which convert energy from the sun into electrical power. It is utilized in various energy sectors, such as PV energy plants and residential PV energy systems. PV energy generation is influenced by various factors, such as the area of solar panels, installation location, season, time of day, solar radiation, and weather conditions. Because solar panels do not emit carbon or pollutants during the energy conversion process, they are considered one of the most environmentally friendly and sustainable energy conversion devices. Additionally, PV energy plants have a minimal impact on the surrounding environment and have low maintenance costs.

PV energy generation must be predicted by considering various factors, and such prediction plays a crucial role in renewable energy conversion and electricity supply planning. The forecasted PV energy generation data are used to generate demand predictions in the energy market to establish energy production plans, which are vital for ensuring stable operation of the energy market and facilitating the transition to renewable energy. Accordingly, this article provides detailed information on the design of PV energy generation prediction models, focusing on aspects such as solar panel capacity, solar position, and weather data to establish approaches for formulating accurate PV energy generation prediction models.

### 3.2.1. PV Capacity according to Area of Virtual Building

This section focuses on the design approach for virtual buildings by considering the inter-relationships between buildings and PV energy generation facilities, with an emphasis on energy production in urban areas and the transition to renewable energy. By designing virtual buildings that are tailored to the intended use of the structure, we explore feasible approaches for predicting PV energy generation capacity in the context of renewable energy conversion. Generally, larger virtual buildings can accommodate greater numbers of solar panels, resulting in higher energy generation. When designing a virtual building, it is crucial to allocate spaces for the installation of solar panels by considering the building's area. Additionally, it is important to adjust the orientation of the installed solar panels by considering the geographical location and climatic conditions of the building to maximize energy generation. This analysis allows us to assess the contribution of virtual building design to renewable energy conversion. By selecting appropriate positions and orientations of solar panels and considering the structural characteristics of the building, the number

of solar panels that can be installed is determined. By accounting for these factors and ensuring fulfillment of the building's energy demand, the PV energy generation capacity of virtual buildings can be determined.

In addition, from the building information modeling (BIM) perspective, we determine the PV energy generation capacity of a building such that the building's area is utilized efficiently by integrating PV energy generation information throughout the building's life cycle. Moreover, by designing a virtual model of a building by using digital methods, the building's components can be supported in a step-by-step manner, thereby facilitating more effective analysis and control compared to manual processes. Virtualization of the design, operation, and management of buildings can enhance the quality and productivity of PV energy generation. By easily understanding and reviewing PV energy facilities, the accuracy and completeness of PV energy generation according to the building's purpose can be ensured while simultaneously deriving optimization strategies for costs and other variables.

Accordingly, to secure the diversity of the area for each virtual building, the area is considered according to four uses. Building area is set differently depending on the usage characteristics of virtual buildings. The building areas of a virtual factory, barn, large building, and small building are 1090.91 m$^2$, 859.52 m$^2$, 330.58 m$^2$, and 178.51 m$^2$, respectively. The area required for the installation of solar panels is limited to less than 70% of the overall building area to ensure safety. The areas in which stable solar facilities can be installed in the designed virtual factory, barn, large building, and small building are 766.94 m$^2$, 601.65 m$^2$, 231.4 m$^2$, and 122.31 m$^2$, respectively. Therefore, considering that the capacity of solar facilities is 3 kW per 23.14 m$^2$ in preparation for the installable area, 99 kW for factories, 72 kW for livestock, 30 kW for large buildings, and 16 kW for small buildings, such systems can be installed and operated.

### 3.2.2. Data Collection Method from the Perspectives of the Sun's Location and Weather Conditions

To design a predictive simulation model, we discuss the data collection and analysis approaches that are closely related to PV energy generation. The data that can be collected to predict PV energy generation include variables such as location, weather conditions, and installation angles and orientations of solar panels. The key variables for predicting PV energy generation can be classified as variables related to the sun's position and variables related to weather conditions.

Data related to the sun's position include the azimuth angle, altitude, and season. The azimuth angle of the sun changes along the horizon with time and location conditions. It is closely related to the solar panel installation angle, and by aligning the panel's installation angle with the azimuth angle of the sun, solar energy generation can be maximized. Altitude directly influences PV energy generation because the amount of solar radiation reaching the solar panels increases as the sun rises higher. Generally, the altitude is lower in the morning and evening, and it is highest around noon. PV energy generation is affected by the location-specific azimuth angle and the sun's altitude. The height and azimuth angle of the sun vary depending on the season, which also affects PV energy generation. For example, in Korea, the sun is lower in winter, resulting in decreased PV energy generation due to a reduction in the amount of direct sunlight that reaches solar panels. In contrast, in summer, the sun is higher, leading to increased PV energy generation because a greater amount of sunlight is directly incident on solar panels. Thus, the sun's azimuth angle and altitude and the season are key variables that affect PV energy generation and are essential for prediction.

Data on variables related to weather conditions are important because they have the greatest impact on PV energy generation and are expressed as solar radiation and cloud quantity. As precipitation increases, more clouds are formed, and PV energy generation decreases, owing to a decrease in solar radiation. In contrast, if cloud cover is scant, the amount of generated PV energy increases. Cloud data are linked, and the values

representing the amount of clouds in the linked data are collected as "sunny", "cloudy", and "lots of cloud". Solar irradiation data indicate the amount of PV energy that is incident on solar panels. Table 1 lists some of the input variable data used to design the simulation model for prediction of PV energy generation.

**Table 1.** Part of the variable database for PV energy prediction simulation design.

| Date | Azimuth | Altitude | Season (Ecliptic Longitude) | Amount of Cloud |
|---|---|---|---|---|
| 28 July 2022 11:00 | 123 | 61 | Major heat (125) | sunny |
| 28 July 2022 12:00 | 152 | 69 | Major heat (125) | sunny |
| 28 July 2022 13:00 | 195 | 70 | Major heat (125) | sunny |
| 28 July 2022 14:00 | 229 | 64 | Major heat (125) | sunny |
| 9 August 2022 11:00 | 128 | 59 | Start of autumn (137) | cloudy |
| 9 August 2022 12:00 | 156 | 66 | Start of autumn (137) | cloudy |
| 9 August 2022 13:00 | 194 | 67 | Start of autumn (137) | cloudy |
| 9 August 2022 14:00 | 225 | 61 | Start of autumn (137) | cloudy |
| 10 January 2023 11:00 | 154 | 26 | Minor cold (289) | sunny |
| 10 January 2023 12:00 | 169 | 29 | Minor cold (289) | sunny |
| 10 January 2023 13:00 | 185 | 30 | Minor cold (289) | sunny |
| 10 January 2023 14:00 | 201 | 27 | Minor cold (289) | sunny |

Data on the solar azimuth angle, solar elevation angle, season, cloud cover, and solar radiation within Seoul are collected and stored in a database to be utilized as variable data in a model for prediction of PV energy generation, such as machine learning algorithms, to ensure accurate prediction.

### 3.2.3. Learning and Testing Data Design Plan

Data standardization is a common preprocessing step before data analysis and modeling. To ensure data reproducibility and obtain consistent results from the same dataset, the data are processed in the same manner. Out of the total 8689 data points included in this study, 6082 data points were assigned to the training set, and 2607 data points were assigned to the test set. The training set was used to train the model, and the test set was used to evaluate the model's performance. The data were divided randomly to prevent the model from learning specific patterns or biases.

Lastly, it is important to determine which type of ML algorithms are the most suitable for the purpose at hand. The data collected herein exhibit the characteristics of time series data. Accordingly, we selected LSTM, which has linear characteristics, as the ML algorithm.

### 3.3. LSTM Model Design

LSTM is a type of RNN that is particularly useful when processing time series sequence data. Figure 2 shows the structure of the LSTM model.

LSTM is used to remember past information and combine it with current inputs to predict upcoming values. It is suitable for analyzing long-term patterns in time series data. Therefore, it is suitable for processing data with irregular time intervals, which are commonly encountered in PV energy generation forecasting. Additionally, LSTM can be useful for predicting future energy generation by considering past weather information and historical energy generation data. It can handle data with various time intervals, and by inputting weather data, it can be trained on historical PV energy generation data in a specific period to forecast future energy generation. Prediction of PV energy generation is influenced by several variables, such as weather conditions, position of solar panels, and

daily fluctuations. These variables must be considered by incorporating both past and current data when building a predictive model.

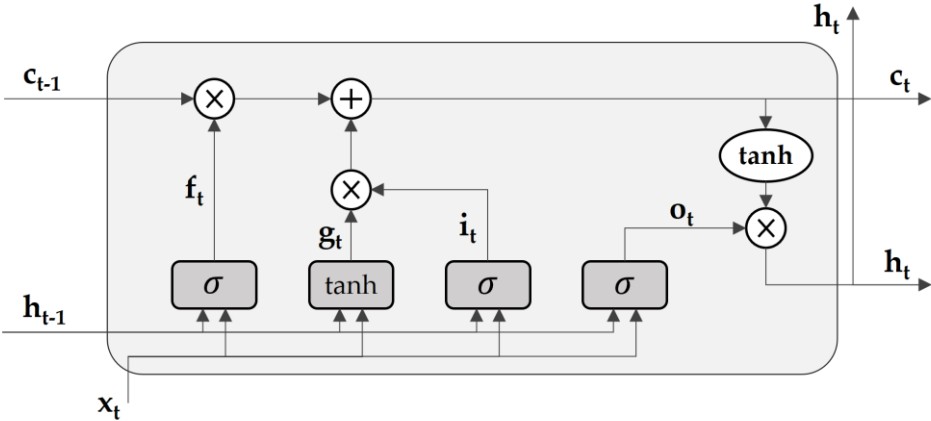

**Figure 2.** Structure of LSTM model.

By using LSTM to predict PV energy generation, accurate prediction results can be obtained. Prediction of solar energy generation plays a crucial role in the operation and management of PV energy plants, and accurate energy generation predictions significantly affect the optimization of power grid operations and formulation of optimal operating plans for energy plants.

### 3.3.1. LSTM Algorithm Design

Figure 3 depicts the structure of the LSTM model, as represented using the forget gate. In Figure 3, the series of computations performed by the forget gate is denoted by sigma nodes.

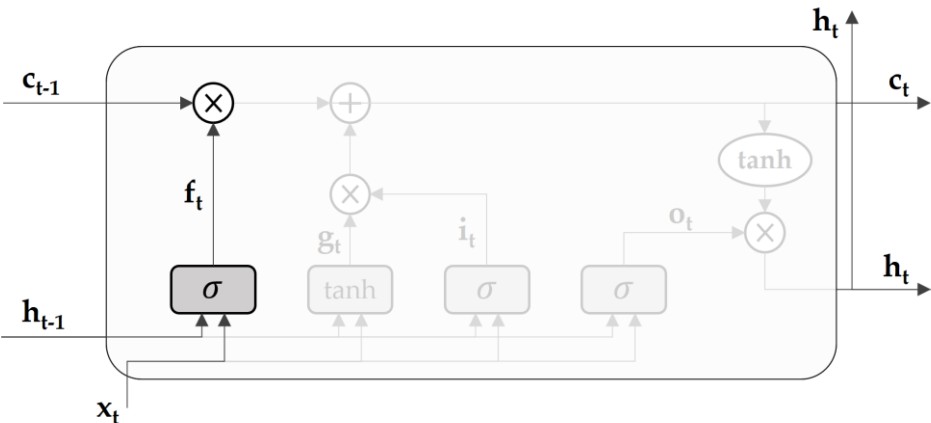

**Figure 3.** Forget gate in structure of LSTM model.

Within a sigma node, there are dedicated weight parameters of the forget gate, which performs the computation expressed in Equation (1). By solving Equation (1), the value output by the forget gate is obtained, which is used to perform element-wise multiplication with the previous memory cell to obtain the result value.

$$f = \sigma\left(x_t W_x^{(f)} + h_{t-1} W_h^{(f)} + b^{(f)}\right) \tag{1}$$

As information is passed through the forget gate, the memories that need to be forgotten are removed from the memory cell of the previous time step. Additionally, a new memory cell is added to store new important information. To incorporate this new information into the memory cell, a tanh node is introduced, as shown in Figure 4.

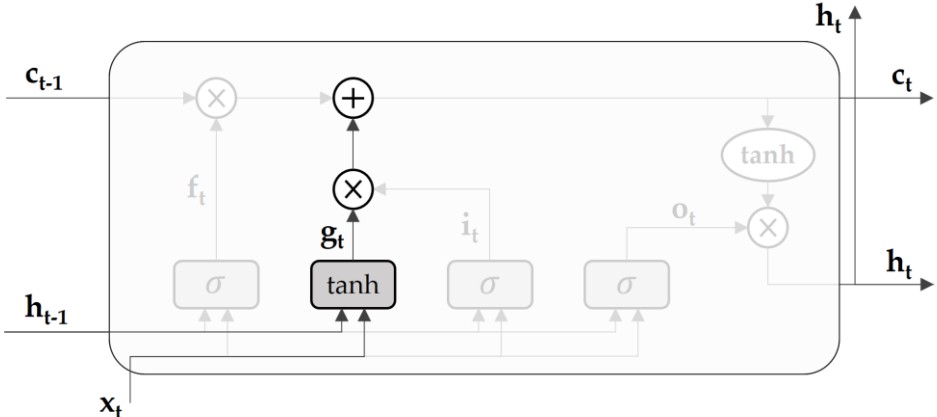

**Figure 4.** New memory cells in the structure of the LSTM model.

The tanh node is not a gate but serves the purpose of adding new information into the memory cell. Therefore, instead of using the sigmoid function, the tanh function is used as the activation function. Equation (2) expresses the addition of new information to the memory cell.

$$g = tanh\left(x_t W_x^{(g)} + h_{t-1} W_h^{(g)} + b^{(g)}\right) \tag{2}$$

By adding an input gate, the computational graph is represented, as depicted in Figure 5. The input gate determines the importance or value of the elements in "*g*" as new additional information.

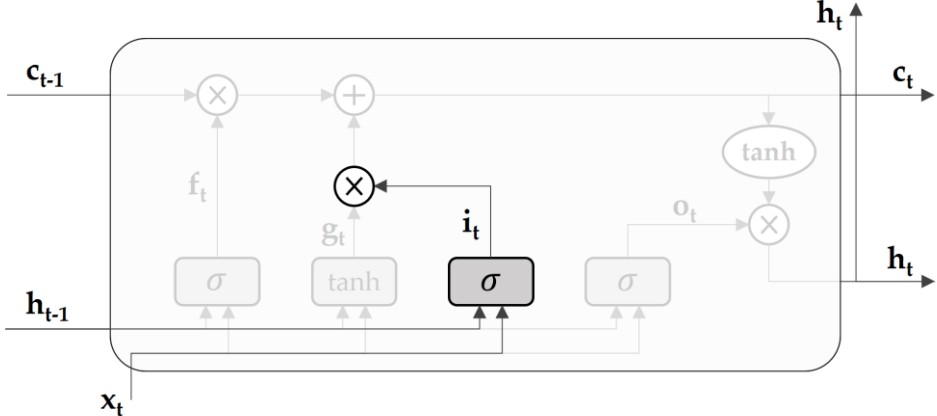

**Figure 5.** Input gate in the structure of the LSTM model.

This gate selectively incorporates new information instead of accepting it blindly. From another perspective, the input gate introduces weighted information. Equation (3) expresses the calculation performed in the input gate. Equations (1)–(3) express the operations carried out within the LSTM model.

$$i = \sigma\left(x_t W_x^{(i)} + h_{t-1} W_h^{(i)} + b^{(i)}\right) \tag{3}$$

It adjusts the importance of each element in determining the next hidden state. Figure 6 depicts the output gate, which is responsible for outputting the next hidden state (*h*).

The state of openness of the output gate is determined from the input (*x*) and the previous state (*h*), the calculation for which is expressed in Equation (4). Note that the weight parameters and biases used here are denoted by adding the first letter of the output (*o*) as a prefix. Similarly, subscripts are added to indicate the gates involved in the process.

$$o = \sigma\left(x_t W_x^{(o)} + h_{t-1} W_h^{(o)} + b^{(o)}\right) \tag{4}$$

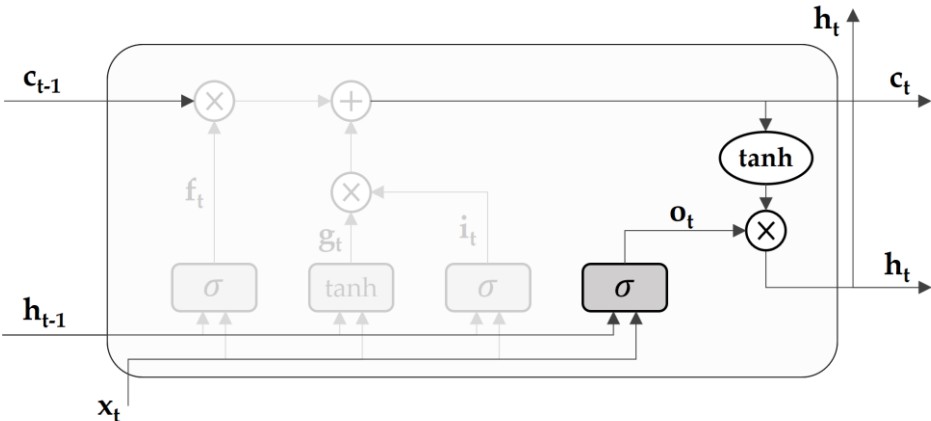

**Figure 6.** Output gate in the structure of the LSTM model.

Equation (4) consists of the input (*x*) multiplied by the weight matrix (*W*) and the previous hidden state (*h*) multiplied by the weight matrix (*W*). The resulting matrices are combined, and the bias term (*b*) is added. Then, the output (*o*) of the output gate is obtained by passing through the sigmoid function. Finally, element-wise multiplication between *o* and *h* is performed to produce the output. Equation (5) expresses a combination of matrix transformations and parallel shifts.

$$c_t = f \odot c_{t-1} + g \odot i \tag{5}$$

The calculation expressed in Equation (6) in the output gate is denoted by the sigma symbol. If we denote the output of sigma as "*o*", *h* is computed by performing element-wise multiplication between *o* and *tanh*(*c*).

$$h_t = o \odot tanh(c_t) \tag{6}$$

### 3.3.2. Prediction Simulation Model Flow

Figure 7 shows the flow of the proposed PV energy generation prediction model, which is divided into data collection, data processing, and model learning. First, data on building infrastructure, solar panel positioning, and weather conditions are collected. Then, the collected data are classified to identify any missing data, and features are extracted from each data point while performing data cleaning. The sampled data are divided into training and test sets for cross validation. Cross validation involves evaluating the model using both the training and test sets and iteratively finetuning the model to avoid overfitting to the test set. Additionally, cross validation helps to prevent data bias and mitigates underfitting due to insufficient data. Finally, the data are processed and used to train a PV energy generation simulation prediction model based on the LSTM algorithm.

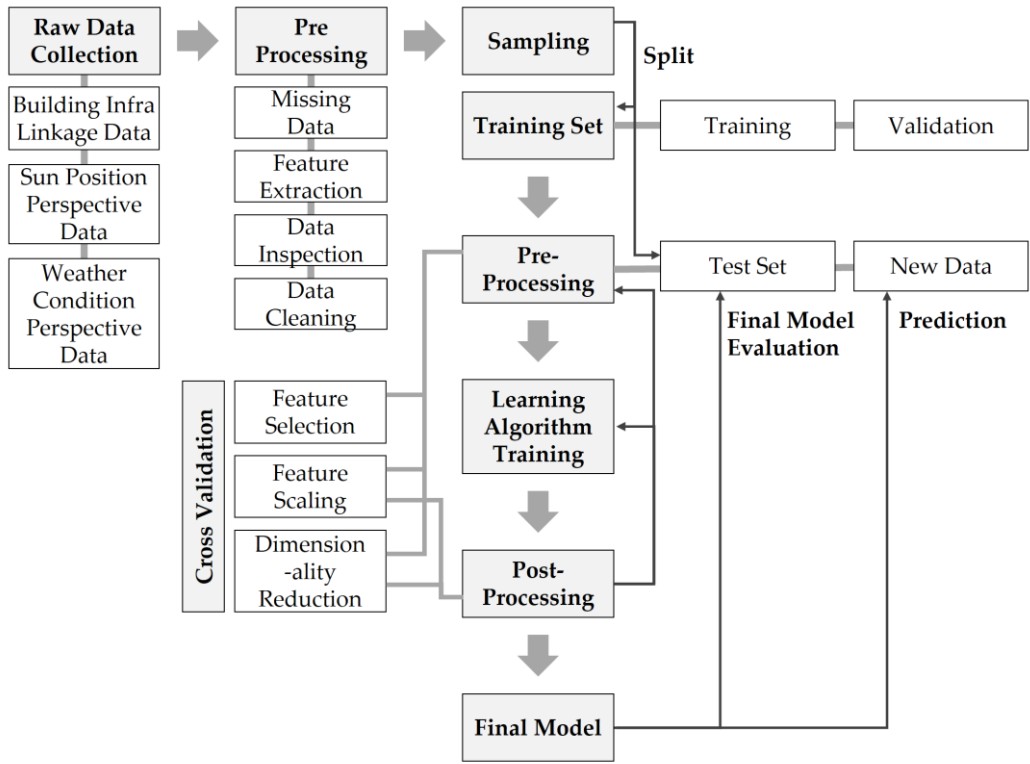

**Figure 7.** Prediction simulation model flow.

## 4. Implementing Predictive Model and Verifying Accuracy

To implement the proposed PV energy generation prediction model, PV energy generation was analyzed, and patterns were analyzed on the basis of linkages between the input variables.

### 4.1. Analysis of Input Variable Data

4.1.1. Point 01: Correlation Analysis between Input Variable Data

Correlation refers to the existence of a linear relationship between two variables, and the analysis of such a relationship is called correlation analysis. If two variables are related, their values tend to change together, and the value of one variable can be estimated from that of the other variable. To understand the relationship between two variables, the data of both variables are collected, and any existing patterns are analyzed.

Figure 8 depicts the scatter plots used to analyze the correlations between the input variables in this study. The X axis represents one variable, while the Y axis represents another variable. However, even if a scatter plot exhibits a specific pattern, statistical verification is necessary because the observed pattern may be attributed to a sample extracted from a population where no relationship exists between the two variables. By using statistics, the strength of such relationships can be determined, and the probability of observing specific patterns in a sample extracted from a population where no relationship exists between the variables can be determined. Thus, Figure 8 confirms a strong positive correlation.

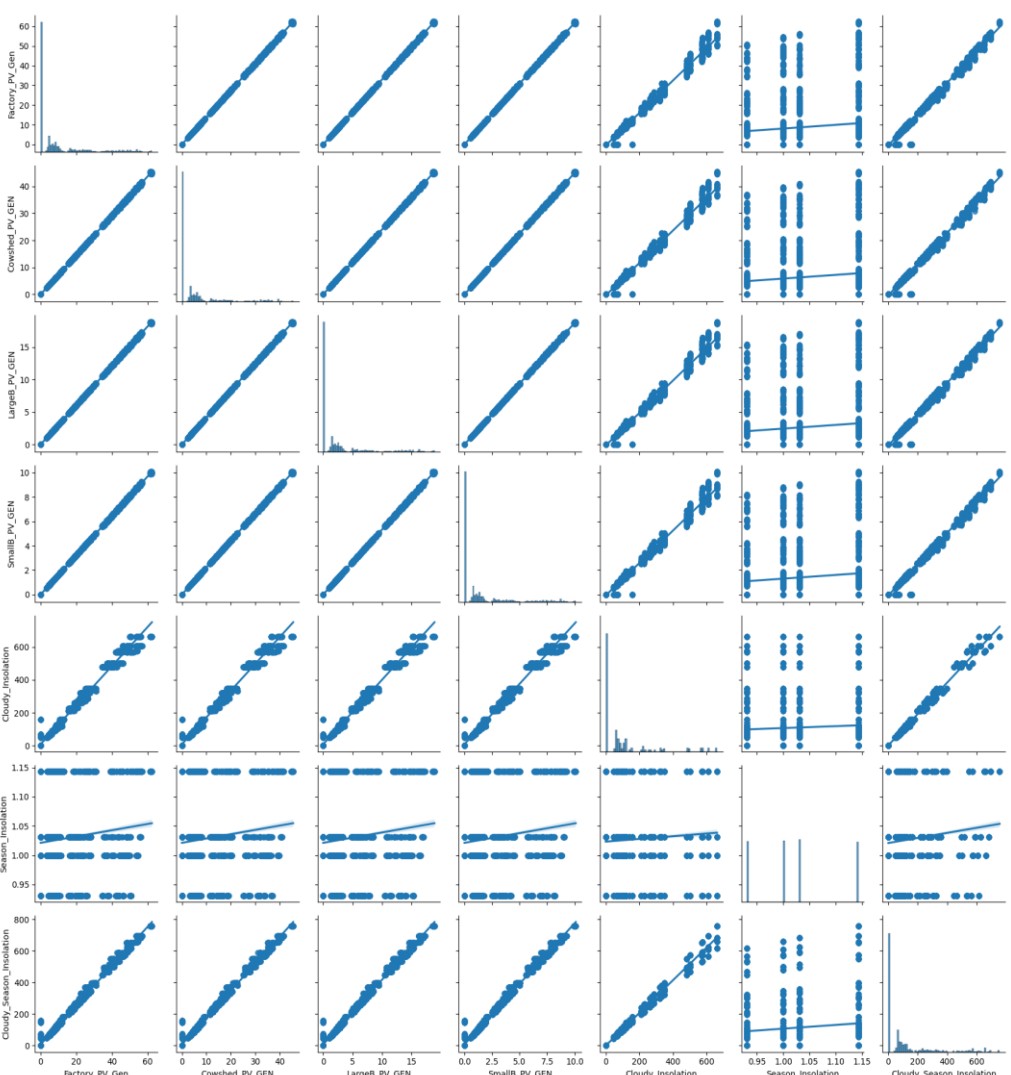

**Figure 8.** Correlation analysis between input variable data.

### 4.1.2. Point 02: PV Energy Generation by Time Zone

The approach for predicting PV energy generation from the perspective of solar position and weather conditions is explained using Equations (7)–(9). The solar energy generation efficiency (SGE) variable is calculated based on the solar facility available area (SFA) and the solar panel installation angle (SIA) using the Equation (7). Relative energy generation (REG) is a variable that considers the perspective of solar panel position. Solar facility available area (SFA), solar installation angle (SIA), and solar panel module area (SMA) are defined as variables linked to the solar panel installation angle and the number of modules that can be installed.

$$\text{SGE} = \frac{\text{REG} \times \text{SFA} \times \cos(\text{SIA})}{\text{SMA}} \tag{7}$$

Seasonal insolation was determined using data provided by the Korea Renewable Energy Data Center, while cloudiness was determined using public data provided by the Korea Meteorological Administration. The formula for calculating insolation is expressed as Equation (8), in which seasonal insolation (SI) is defined as a function of the cloudy variable.

$$\text{Insolation} = \text{SI} \times \text{Cloudy Variable} \tag{8}$$

Equation (9), which is based on Equations (7) and (8), is used to calculate PV energy generation (PV_GEN).

$$PV\_GEN = Insolation \times SGE \times Module\ Efficiency \tag{9}$$

Figure 9 depicts the PV energy generated in a specific season on buildings with different PV energy generation facilities. According to the figure, the energy generation window is from 0800 to 1900, and the peak PV energy generation period is from 1100 to 1400.

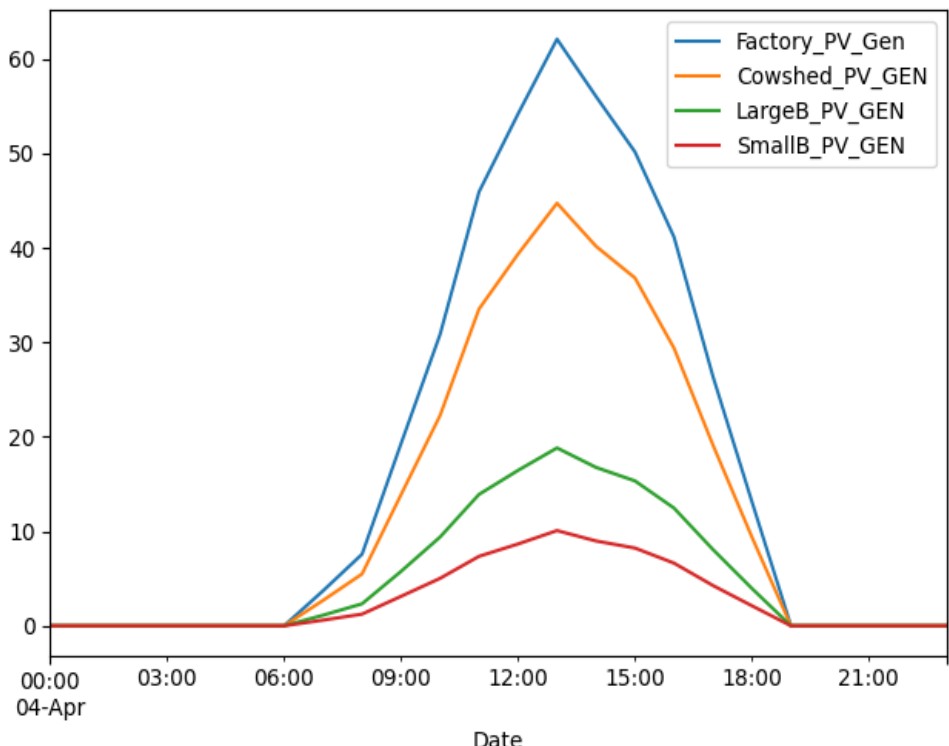

**Figure 9.** PV energy generation by time according to building type.

### 4.1.3. Point 03: Analysis and Results of PV Energy Generation by Season

Figure 10 depicts the monthly total PV energy generation from the perspectives of the sun's position and seasons. It represents the variation in energy generation over the course of 12 months, highlighting the notable points in each season.

- Insights for spring (March to May): There were fewer instances of rainfall and less cloud cover, resulting in prolonged clear days. As a result, the total PV energy generation was highest between March and May.
- Insights for Summer (June to August): Various climatic factors, such as the number of rainy days, rainfall amount, and cloud cover, influenced solar energy generation. Owing to a relatively long monsoon period and fewer clear days, solar radiation was insufficient, leading to lower power generation during summer.
- Insights for autumn (September to November): September and November recorded similar energy generation levels. This season benefitted from moderate temperatures, high solar radiation, and clear weather, resulting in efficient PV energy generation.
- Insights for winter (December to February): With shorter daylight hours and cloud cover associated with precipitation (snowfall), PV energy generation decreased during this season.

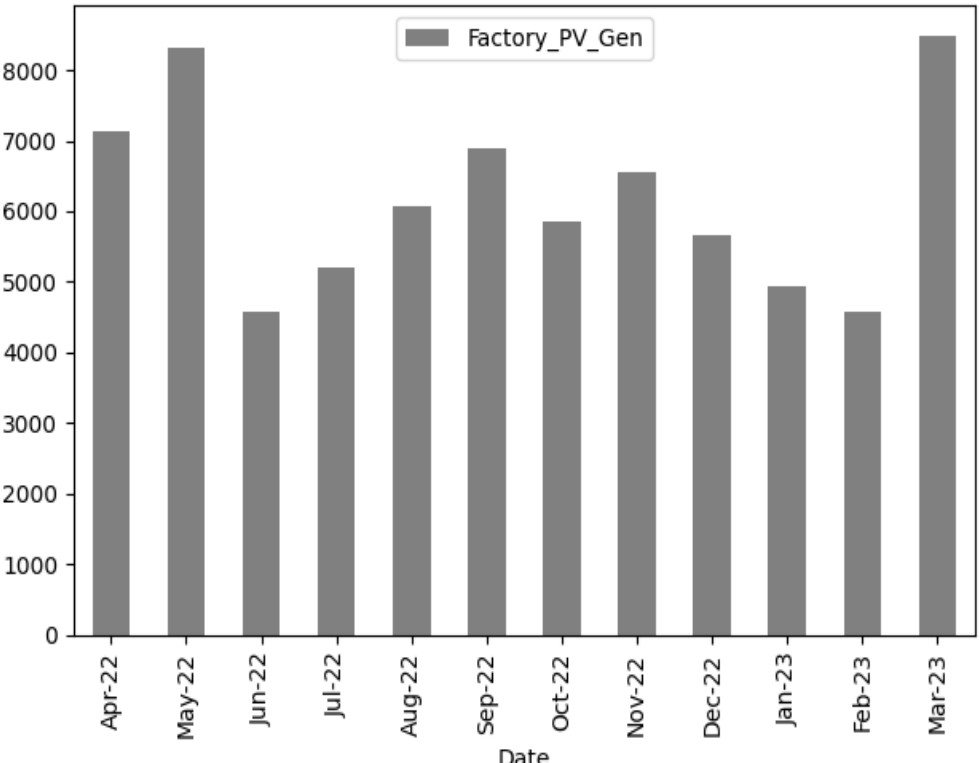

**Figure 10.** Seasonal PV energy generation in factory buildings.

Comparing according to season shows that the amount of generated PV energy descends in the following order: spring, autumn, summer, and winter.

### 4.1.4. Point 04: Analysis of PV Energy Generation according to Amount

Since the amount of solar radiation received from the sun decreases when cloud cover is high, the amount of energy generated was analyzed considering the meteorological condition variable. For example, in the presence of thick clouds, sunlight is blocked, and energy generation is reduced.

As a result, the amount of energy generated can be affected by the type of cloud. The amount of PV energy generated depending on the change in the amount of cloud cover over time is depicted in Figure 11. When the sky is clear, the amount of energy produced is the highest, followed by the cloudy and many clouds conditions, generating a large amount of PV energy. Under the same conditions, when comparing and analyzing PV energy generation according to the extent of cloud cover, PV energy generation occurs in descending order as follows: sunny, cloudy, and lots of clouds.

### 4.1.5. Point 05: Analysis of Time, Transportation, and PV Energy Data

Figure 12 illustrates the interconnected patterns among time, cloud cover, and PV energy generation in a representative virtual building, namely a factory. It depicts the analysis of the influences of time, solar position, and cloud cover on energy generation in a three-dimensional context. When examining the data over a one-year period, it can be observed that PV energy generation is relatively higher in environments without clouds compared to that in environments with different cloud cover conditions. Therefore, the order of superior PV energy generation, regardless of the season, is as follows: clear skies, partial cloudiness, and heavy cloud cover. This indicates that PV energy generation varies depending on the amount of cloud cover, regardless of the season.

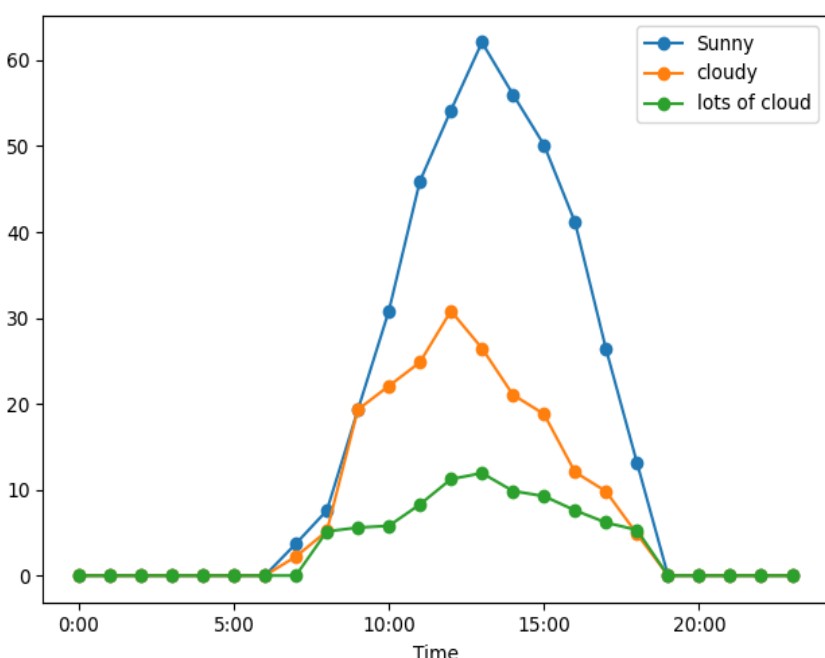

**Figure 11.** Amount of energy generated by change of cloud.

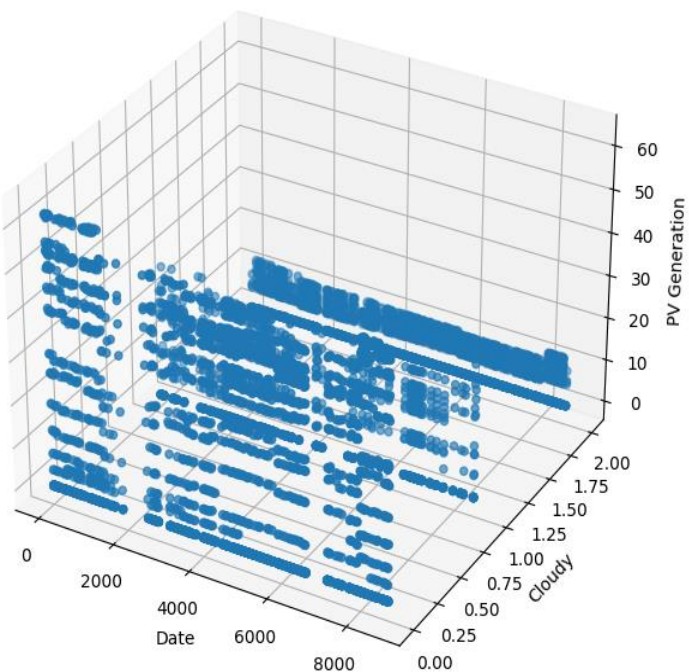

**Figure 12.** Correlation analysis in 3D space.

### 4.2. LSTM Model Prediction Result Based on PV Energy Generation

By removing missing values from the entire dataset, diversity was ensured, and based on the PV energy generation data at the building level, it was possible to predict future PV energy generation. Figure 13 depicts the division of the dataset into the training and test sets with a certain ratio before predicting PV energy generation for a virtual building using LSTM.

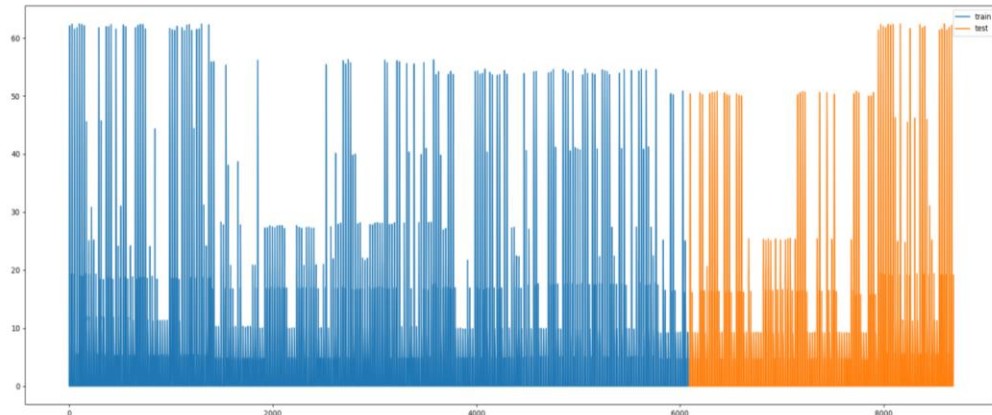

**Figure 13.** Remove Missing Values in Data and Classify Data Sets.

Figure 14 depicts the result of the PV energy generation prediction model at a specific time; the loss was attributed to an error of 0.007 between the actual value and the value predicted by the model. PV energy generation was predicted by focusing on the season, cloud cover, solar radiation, and the sun's location. The blue graph in Figure 14 shows the actual data values, and the orange graph shows the predicted values.

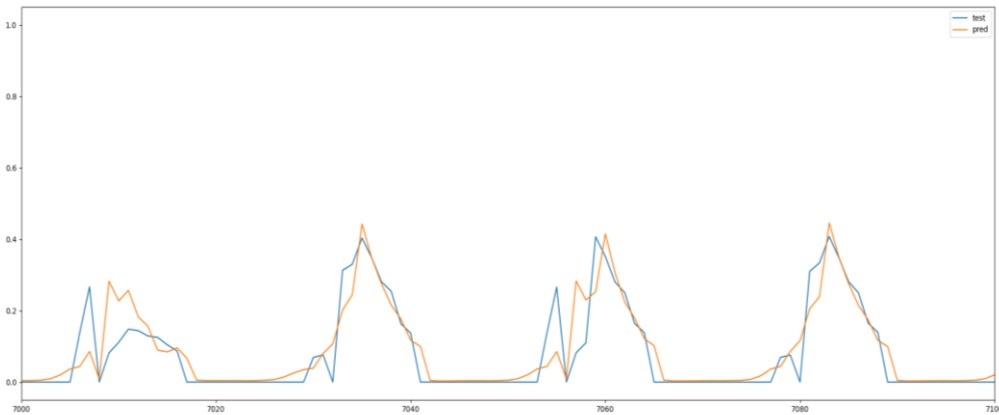

**Figure 14.** Energy generation prediction model result for a particular section.

An analysis of the predictions of PV energy generation indicated that the predictions generated for sunlight hours were relatively less accurate than those generated for the sunset hour, owing to the change in the amount of clouds at the time of sunrise. At the time of sunset, solar radiation was constant, without much change; therefore, it was possible to predict PV energy generation with an accuracy exceeding 90%.

## 5. Carbon-Neutrality Based on PV Energy Prediction Model and EV Charging Platform

Integrating EV charging platforms with PV energy generation prediction analysis plays a crucial role in pursuing sustainable energy and carbon neutrality.

### 5.1. PV Energy Prediction Model and EV Charging Platform Integrated Simulation

Enhancing the integration between PV energy generation prediction models and EV charging platforms is crucial to facilitate a high level of PV energy supply to the chargers during periods of energy availability at the building level and effectively manage energy consumption. When integrating a PV energy generation prediction model with an EV charging platform, EV charging can be adjusted during times when PV energy generation is high in terms of EV charging scheduling to induce direct use of energy generated by PV. Through the integration of a PV energy generation prediction model with EV charging

platforms, users can plan charging during high-PV-energy-generation periods, providing information on charging with renewable energy sources to support eco-friendly charging practices. Additionally, from the perspective of the renewable energy market, encouraging EV charging during low electricity tariff periods helps to avoid charging during high-carbon-emission periods. This approach allows for increased utilization of renewable energy and reduces carbon footprint. By integrating a PV energy generation prediction model with EV charging platforms, we can ensure the stability of renewable-energy-based grids and contribute to the development of a sustainable EV ecosystem. To achieve this, we conducted three inter-related simulations aimed at securing the stability of renewable-energy-based grids, supporting the sustainable development of the EV ecosystem, and reducing carbon emissions.

The establishment of an interconnected system through comprehensive digitization of infrastructure and energy sources on the EV charging platform is paramount. An integrated EV charging platform, linking EV charging infrastructure with PV energy generation, assists in managing energy supply based on charging demands. Bidirectional data sharing between energy supply and EV charging infrastructure demands enhances energy and infrastructure utilization efficiency. Users can monitor the status and usage of EV chargers installed within buildings, ensuring the reliability of the PV energy supply system based on charging demands through the interconnected EV charging platform. This seamless integration secures the reliability of EV charging and maximizes energy utilization and infrastructure efficiency from both energy supply and EV charging infrastructure perspectives.

Figure 15 presents three approaches for integrating the PV energy generation prediction model with EV charging platforms: a PV-EV charging infrastructure integration simulation, EV charging control, and PV energy generation sharing. The simulation is based on a charging scenario for EVs, assuming a charging fee of USD 0.3 per 1 kWh. For a full EV charge, 12 h charging session was considered with continuous charging at a rate of 7 kW per hour.

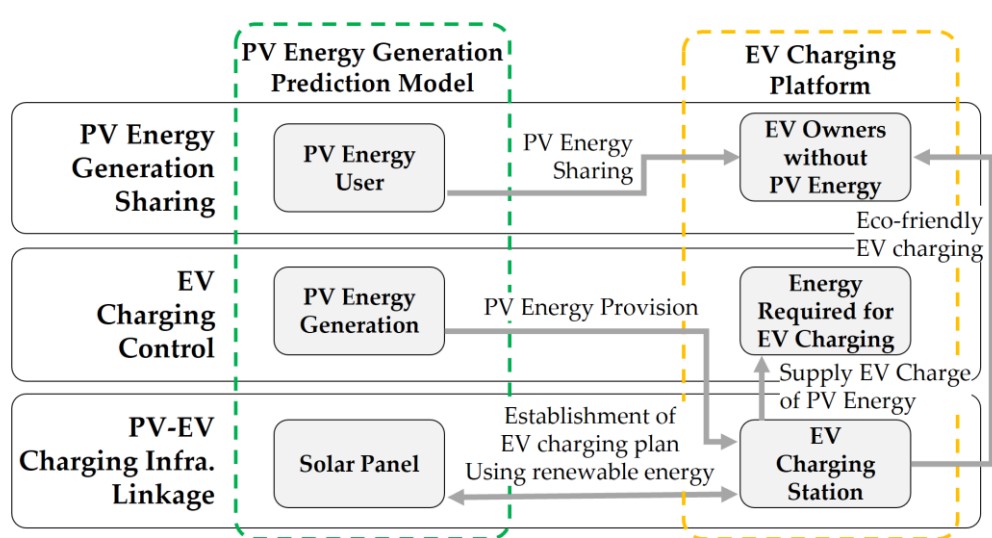

**Figure 15.** PV energy generation prediction model—EV charging platform integrated method.

- PV-EV charging infrastructure integrated simulation: Users can plan their EV charging based on PV energy generation. When solar PV generation is ample, an eco-friendly EV charging ecosystem is established during electric vehicle charging. This initiative aims to secure energy source diversity, thereby expanding energy self-sufficiency and utilization. Simulation incorporates data on the usage frequency of chargers within the EV charging platform, available charging hours, charging loads, and charging times to conduct the analysis. Within a large building, the daily maximum PV energy

generation is 124.4 kW, allowing for the simultaneous charging of approximately 1.48 EVs.

- EV charging control based on PV energy generation: Optimization of energy utilization is achieved by adjusting both PV energy generation and electric vehicle charging. This approach aims to reduce the grid load and enhance the economic viability of PV generation. Simulation involves controlling charging based on data such as the usage frequency of chargers within the EV charging platform, charging loads during EV charging, charging fees according to charging type at different times, and charger status data. By supplying the grid with the PV energy (124.4 kW) that can charge approximately 1.48 EVs in a large building, it is possible to save USD 37.32 in fees.
- PV energy generation sharing: Users share generated PV energy to facilitate eco-friendly EV charging, allowing non-self-generating users to contribute to carbon-neutral ESG goals. Leveraging data such as the location of charging stations within the EV charging platform, the number of available chargers, and charging start/end times, users can share PV energy generation among themselves. During EV charging, this sharing mechanism reduces $0.63(\text{tCO}_2\text{eq}/\text{kW})$ of carbon emissions per 1 kW capacity, contributing to the achievement of carbon neutrality.

Therefore, simulation results from 3 perspectives demonstrate the establishment of an eco-friendly EV charging ecosystem, reduced grid load, and carbon reduction effects. By supplying balanced energy, the simulation ensures the stability of charging demand. This approach contributes to climate change mitigation; guarantees a sustainable EV charging system; and presents a new need and future vision for infrastructure expansion, energy demand–supply, and eco-friendly models.

*5.2. Analysis of Carbon Footprint Result*

To establish a carbon reduction model in view of future energy supply, the amount of carbon emitted during PV energy generation and reduction measures thereof should be analyzed. The carbon emission source unit of the carbon reduction technology is $0.63(\text{tCO}_2\text{eq}/\text{kW})$, and carbon reduction is derived using the energy generation and reduction source units, as expressed in Equation (10).

$$\text{Carbon Emission}(\text{tCO}_2\text{eq}) = 0.63(\text{tCO}_2\text{eq}/\text{kW}) \times \text{Facility Capacity (kW)} \quad (10)$$

Typically, when a factory building produces 410 kW of energy in a day and the capacity of the PV energy facility installed in the building is 99 kW, the amount of carbon generated by the building can be reduced by $62.37(\text{tCO}_2\text{eq})$ per year. If the life of the solar plant is set to 25 years, the total $CO_2$ reduction during the plant's operations is $1559(\text{t O}_2\text{eq})$. This carbon reduction value was compared to the amount of carbon absorbed by pine trees. Considering the average annual $CO_2$ absorption of a pine tree, as reported by the National Forest Research Institute, the reduction effect of PV energy generation in the factory is equivalent to 26,540 pine trees. Therefore, efficient use, storage, and operation of energy from the perspective of energy supply can reduce carbon emissions substantially compared to those associated with conventional energy use.

## 6. Conclusions and Discussion

In this paper, we focused on a model for predicting PV energy generation integrated with EV platforms. Planned transition towards carbon-neutral ESG, renewable energy conversion for buildings, and widespread EV adoption aims to strategically utilize PV energy generation for carbon reduction and energy supply to EV charging infrastructure. Although ML-based models for PV energy generation prediction have been reported, it is challenging to extend them as simulation models for analyzing carbon reduction and EV platform linkage. To solve this problem, we proposed a plan to link EV charging platforms and carbon reduction measures in terms of renewable energy conversion using a solar power generation prediction model that combines data and artificial intelligence.

The proposed model allows users to use PV energy from the perspective of transition to renewable energy without compromising their existing energy usage patterns. In this manner, users can reduce carbon emissions. Additionally, our proposal includes the integration of PV-EV charging infrastructure through a PV energy generation prediction model and EV charging platform. The simulation allows for EV charging control and sharing of energy generation. With the predictive model, users can plan their charging during high-PV-energy-generation periods, and the EV charging platform provides information to enable eco-friendly charging using renewable energy sources. From a city planning perspective, with the proposed approach, it becomes possible to design models for renewable energy transition and carbon reduction in the initial stages of city planning by using only data related to building areas, solar positions, and weather conditions.

Research was conducted with a focus on building-level data. However, future research will expand the scope by incorporating PV energy generation data at regional hubs and analyzing carbon reduction potential based on EV charging demands at the city level. This will explore approaches to stabilize renewable energy supply and achieve carbon-neutrality. Ultimately, the global EV market is expected to continue growing, highlighting the need for further research on PV energy generation prediction models and the integration of EV charging platforms to achieve carbon-neutral ESG goals. Extending the scope and conducting future research in this area will be essential. Continuing research based on the PV energy generation prediction model and its integration with EV platforms from a carbon-neutral perspective will lead to the development of a data-centric smart city. Applying ESG will help industries adapt to environmental climate change, manage carbon emissions, and achieve carbon neutrality.

Through such innovative research, addressing climate change from building- to city-level perspectives, the energy supply–demand imbalance arising from regional differences in energy generation and consumption can be tackled. Additionally, to address energy supply–demand challenges at the local level, further research should be conducted to develop specialized models based on PV energy generation prediction and EV charging platform integration, considering the specific attributes of each domain within the city, with buildings as the focal point. This approach aims to create tailored solutions that align with the unique energy demands and characteristics of different areas within the city.

Through value-chain-based social market competitiveness and a new EV platform market focused on sustainability, a green strategy is established. As a result, by utilizing PV energy generation for carbon reduction, it is anticipated that a carbon-neutral city can be established from the perspectives of energy security and self-sufficiency.

**Author Contributions:** Conceptualization, S.P. (Sehyun Park), G.Y. and S.K.; data curation, G.Y., S.K., H.S., K.C., T.L. and J.P.; methodology, S.P. (Sehyun Park), D.S., G.Y., K.C., M.-i.C., B.K. and S.P. (Sangmin Park); software, G.Y., S.K., H.S., H.J. (Hyeonwoo Jang), S.L. and H.J. (Hyeyoon Jung); project administration, S.P. (Sehyun Park); visualization, G.Y.; supervision, S.P. (Sehyun Park); validation, G.Y. and S.K.; writing—original draft, G.Y., S.K., H.S. and S.P. (Sangmin Park); writing—review and editing, G.Y. All authors have read and agreed to the published version of the manuscript.

**Funding:** This work was supported by the Human Resources Development (No. RS-2023-00244347) of the Korea Institute of Energy Technology Evaluation and Planning (KETEP) grant funded by the Korea government Ministry of Trade, Industry and Energy, and this research was supported by the Chung-Ang University Research Scholarship Grants in 2021.

**Data Availability Statement:** Data are contained within the article.

**Conflicts of Interest:** The authors declare no conflict of interest.

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
