# Peer review of "Carbon-Neutral ESG Method Based on PV Energy Generation Prediction Model in Buildings for EV Charging Platform"

_buildings, doi:10.3390/buildings13082098_

Round 1
Reviewer 1 Report
Dear authors, the research is rather interesting and presents essential practical results. The minor comments are the following: (1) to clarify goal of the investigation and methods in Abstract, (2) to add Discussion section to discuss the results, key limitations and further prospects
Author Response
We thank the reviewer for their valuable comments and encouragement, and we greatly appreciate their time and effort in going through our manuscript. We fully agree with the reviewer’s comments. Therefore, this manuscript has been revised as follows.
Please see the attachment. Thank you.

Reviewer 2 Report
Dear Authors,
The idea proposed in the paper is interesting and worth exploring. Following are my queries to improve the paper further.
What is the reason for the adoption of LSTM? If possible, compare with other machine learning algorithms depicting the advantages of LSTM.
Please explain how the data for solar PV was obtained.
The advantage and application of the EV charging platform is not clear. Please provide an additional explanation with supporting data or simulation results for the EV charging platform.
What is the relationship between predicting PV output and the EV charging platform?
Briefly explain the future work in the conclusions.
The English language is fine.
Authors are suggested to review the paper thoroughly to eliminate minor grammatical mistakes.
Author Response

(The authors gave the same response as above.)

Reviewer 3 Report
- The abstract must be revised and improved.
- The Introduction / background section also needs to be improved with latest research work proposed in this research area since last 3 years.
- Authors should mention clearly about the novelty of the paper and contributions in the introductory section and in the abstract.
- Methodology section is well explained and easy for the reader to follow
Author Response

(The authors gave the same response as above.)
